# Rostrocaudal Distribution of the C-Fos-Immunopositive Spinal Network Defined by Muscle Activity during Locomotion

**DOI:** 10.3390/brainsci11010069

**Published:** 2021-01-07

**Authors:** Natalia Merkulyeva, Vsevolod Lyakhovetskii, Aleksandr Veshchitskii, Oleg Gorskii, Pavel Musienko

**Affiliations:** 1Institute of Translational Biomedicine, Saint-Petersburg State University, Universitetskaya emb., 7-9, 199034 Saint-Petersburg, Russia; gorskijoleg@gmail.com; 2Pavlov Institute of Physiology RAS, Makarov emb., 6, 199034 Saint-Petersburg, Russia; v_la2002@mail.ru (V.L.); veschickiyalex@mail.ru (A.V.); 3Children’s Surgery and Orthopedic Clinic, Department of Nonpulmonary Tuberculosis, Institute of Physiopulmonology, Politekhnicheskaya ul. 32, 194064 Saint-Petersburg, Russia

**Keywords:** decerebrated cat, backward and forward stepping, C-Fos technique, locomotor networks, spinal cord

## Abstract

The optimization of multisystem neurorehabilitation protocols including electrical spinal cord stimulation and multi-directional tasks training require understanding of underlying circuits mechanisms and distribution of the neuronal network over the spinal cord. In this study we compared the locomotor activity during forward and backward stepping in eighteen adult decerebrated cats. Interneuronal spinal networks responsible for forward and backward stepping were visualized using the C-Fos technique. A bi-modal rostrocaudal distribution of C-Fos-immunopositive neurons over the lumbosacral spinal cord (peaks in the L4/L5 and L6/S1 segments) was revealed. These patterns were compared with motoneuronal pools using Vanderhorst and Holstege scheme; the location of the first peak was correspondent to the motoneurons of the hip flexors and knee extensors, an inter-peak drop was presumably attributed to the motoneurons controlling the adductor muscles. Both were better expressed in cats stepping forward and in parallel, electromyographic (EMG) activity of the hip flexor and knee extensors was higher, while EMG activity of the adductor was lower, during this locomotor mode. On the basis of the present data, which showed greater activity of the adductor muscles and the attributed interneuronal spinal network during backward stepping and according with data about greater demands on postural control systems during backward locomotion, we suppose that the locomotor networks for movements in opposite directions are at least partially different.

## 1. Introduction

Most vertebrates are capable of performing multiple forms of locomotion, in particular, forward (FW) and backward (BW) stepping. Locomotion is based upon the activity of multiple neuronal networks located in the brain and spinal cord. Spinal locomotor networks generating the rhythmic activity of the hindlimb muscles are central pattern generators leading to motoneuronal activity and determining basic motor patterns [1,2]. Contrary to the extensive knowledge about the neuronal mechanisms controlling FW stepping, less data has been collected about BW stepping. Herewith, the former has been getting popular in the light of elaboration on an effective strategy for rehabilitation [3,4,5]. Up to today, the question of whether locomotion in non-FW directions shares the same spinal locomotor networks with FW stepping is unresolved [6,7,8,9].

One powerful technique for both the investigation of the basic locomotor mechanism and motor function improvement is epidural stimulation (ES) [10]. We have shown earlier that epidural stimulation (ES) of the spinal cord is efficient to trigger locomotor circuitry in decerebrated cats—a well-used model in motor studies [11,12]. The neuronal networks responsible for FW locomotion are distributed broadly in the lumbosacral spinal cord and can be activated by ES of any of the L3-S2 segments. On the contrary, only ES of a limited zone around the L6 segment could activate the networks generating BW locomotion [13].

The *c-fos* gene belongs to the immediate early genes family and its expression pattern is a powerful tool for the mapping of interneuronal networks activated during locomotor tasks [14,15,16]. Using this method, we have found a particular difference in the distribution of spinal FW- and BW-controlling networks at the transverse plane of the L6-L7 segments [13] and within the intermediate grey matter, a significantly higher number of C-Fos–immunopositive (FOS+) interneurons was revealed in BW-stepping cats compared with FW-stepping cats.

The aim of the present study was to investigate the rostrocaudal distribution of the spinal locomotor networks in the lumbosacral segments activated during FW and BW stepping using the C-Fos technique and the subsequent comparison of the FOS+ labeling with the location of the motoneuronal pools controlling the different groups of muscles [17,18] and the electromyographic (EMG) activity of the hindlimb muscles. In the previous paper only rude segmental distribution was documented and no links with the motor outputs during BW and FW stepping was provided. The results have significant insights into circuitry-level mechanisms that underlie multi-directional training and site-specific neuromodulation by epidural spinal cord stimulation.

## 2. Methods

*Subjects.* Eighteen normal pigmented adult cats (*Felis catus*) weighing 2.5–3.5 kg were divided into two groups: 6 cats for the C-Fos study (“C-Fos group”) and 12 cats for the EMG activity study (“EMG group”). The number of animals used for the EMG analysis were chosen based on power calculations (G*Power 3.1.9.7 program [19]). All experimental procedures were conducted in accordance with a protocol approved by the Animal Care Committee of the Pavlov Institute of Physiology, St. Petersburg, Russia and followed the European Community Council Directive (2010/63EU) and the guidelines of the National Institute of Health Guide for the Care and Use of Laboratory Animals, Animal Welfare Assurance #A5952-01. Animals were reared in special breeding colony rooms with a 12 h/12 h light/dark cycle and were provided with food and fresh water *ad libitum*. The animal rooms were served by two technicians and veterinarians of the Pavlov Institute Veterinary Department. Care of the animals was conducted on a daily routine basis and medical consultation was provided if needed. Before the acute experiment, veterinarians inspected all animals and only fully healthy animals were used. All experiments were carried out in the morning. All experimental cats were also used for other investigations related to either EMG recordings or C-Fos analysis [13] For all tests, only animals able to perform both BW and FW stepping were used. Individual animals were used and analyzed randomly.

*Surgical procedures.* Experimental procedures and reasons for the use of any protocols were described previously [12,13]. In brief, cats were decerebrated at the precollicular-postmammilar level under deep isoflurane (2–4%) anesthesia. A laminectomy was performed in the lumbar area. An EMG recording was carried out using bipolar EMG electrodes (0.2 mm flexible stainless-steel Teflon-insulated wires; AS632, Cooner Wire, Chatsworth, CA, USA) were implanted [12,20] into the *m. iliopsoas* (IP; *n* = 12), *m. rectus femoris* (RF; *n* = 6), *m. vastus medialis* (VM; *n* = 6) and *m. adductor magnus* (ADD; *n* = 12). Anesthesia was discontinued after the surgical procedures and the experiments were started 2–3 h thereafter. During the experiment, the rectal temperature, electrocardiography and breathing rate of the animals were monitored. The signals from the EMG electrodes were differentially amplified (bandwidth of 10 Hz to 5 kHz; model 1700 Differential AC Amplifier, A-M Systems, Sequim, WA, USA), digitized at 20 kHz with an A/D board (LTR-EU-16, LTR11, L-Card, Moscow, Russia) and processed using computer programs.

*EMG recording.* The head of the decerebrated animal, the vertebral column and the pelvis were fixed in a rigid frame (Figure 1A). The hindlimbs were positioned on the treadmill. The distance between the treadmill belt and the fixed pelvis was 21–25 cm. To evoke locomotion by ES, a ball electrode (d = 0.5 mm) was positioned on the *dura mater* of the dorsal surface of the spinal cord at the L5–L7 segments; this site was used since we had previously obtained a local locus of the spinal cord dorsal surface where ES was able to induce BW stepping [13]. We used the following parameters of stimulation: frequency, 5 Hz; pulse duration, 0.2–0.5 ms; current, 80–300 μA. To evoke FW and BW locomotion, backward and forward motion of the treadmill belt (in relation to the cat) was used [12]. For the EMG study, two successful replications of the locomotor tests were recorded. For the test with hindlimb adduction (“abduction test,” see Discussion), the same stepping was evoked but one limb was abducted 4 cm from the treadmill midline by a thin rope gently turned around the animal’s ankle (this test was performed only for 4 animals from the EMG group). In every cat, for all tests, anterior/posterior (A-P) limb movements were estimated by the precision single-turn potentiometer sensor and recorded synchronously with the EMG signals. The stability of locomotor movements of each individual limb was estimated using the self-similarity coefficient (the amplitude of the autocorrelation function’s second peak in the time series of A-P movements of each individual limb). To characterize step width, reflective markers were placed on the lateral malleolus and the rear view of the walking cat was video recorded (50 frames/s; shutter controlled by external synchronization signal; daA1280-54uc, Basler AG, Ahrensburg, Germany) and analyzed frame by frame. Integrated EMG activity was analyzed to evaluate the differences between the FW and BW stepping. A step period was subdivided into 20 sub-periods (10 during stance and 10 during swing) and EMG activity was measured in each sub-period. Averaged curves for cats stepping BW and FW were compared. The selection of BW and FW trials for further EMG analysis was performed by two experts.

*Kinematic analysis*. To test kinematic, reflective markers were placed on the iliac crest, femoral head, lateral condyle of the femur, lateral malleolus and the fifth metatarsal joint and the side view of the cat was video recorded (60 frames/s). Thereafter, video was analyzed frame-by frame and joints angles were calculated at the moment when the limb was maximally flexed and maximally extended during the swing and the stance step phase respectively [12].

*C-Fos experiment design.* In the 2–3 h after surgery, during which the decerebrated cats recovered from anesthesia, the ability of each individual cat to perform FW or BW locomotion in response to ES of different spinal segments was briefly (during 3–5 min) tested. Six cats that had quite stable locomotion were used for the C-Fos labelling of neurons activated during long-lasted ES-evoked FW (cats Fw1, Fw2 and Fw3) or BW (cats Bw1, Bw2 and Bw3) locomotion [13]. The specific locomotor mode for the C-Fos experiment (FW or BW stepping) was chosen randomly. For 1.5–2 h [21,22], the cats performed 1–2 min of locomotion alternated with 2–4 min of rest.

*Histological protocol.* At the end of the experiments, the animals were deeply anaesthetized with isoflurane (5%) and then perfused transcardially with 0.9% NaCl (2.0 L) in 0.1 M phosphate-buffered saline (PBS) at pH 7.4, followed by 4% paraformaldehyde (2.0 L) in 0.1 M PBS at pH 7.4. The lumbosacral cord was divided into segments based upon the grouping of the dorsal rootlets [23]. Equally spaced 50 μm transverse slices were processed for immunohistochemical protocol to label FOS+ neurons (5 slices per segment). Slices were processed as free floating. A protocol for the immunostaining was previously published [13]. After antigen unmasking, endogenous peroxidase activity blocking and non-specific staining, slices were incubated for 70 h in polyclonal rabbit primary antibodies to C-Fos (PC38-100U, Calbiochem, San Diego, USA; AB_2106755; 1:10,000 dilution). Then, slices were incubated in secondary antibody (biotinylated goat anti-rabbit IgG, BA-1000; Vector Laboratories, Burlingame, USA; 1:600 dilution) for 24 h, followed by incubation in avidin-biotin horseradish-peroxidase complex (ABC Elite system; Vector Laboratories, Burlingame, USA) for 1 h and reacted with diaminobenzidine. After washing in distilled water, sections were mounted, dehydrated, cleared and placed under coverslips. Histological images were analyzed with a microscope (Olympus-CX33; Olympus Corporation, Tokyo, Japan) equipped with a camera Nikon D3400 (Nikon Corporation, Tokyo, Japan). FOS+ neurons were visualized due to the nuclear staining, which looked like a grey or black round-to-oval-shaped loci (Figure 1A). For the rostrocaudal distribution analysis, 40 slices in total were used over the L1-S1 segments. An analysis of the histological slices was performed by two experts. FOS+ neurons were counted across the entire slice area including dorsal and ventral horns and intermediate grey matter.

*Statistical analysis*. Two types of data analysis (EMG data or C-Fos data) were assessed independently by different investigators. The primary outcome was an unambiguous detection of FOS+ neurons within the spinal cord grey matter and a successful registration of the EMG activity of the hindlimb muscles. Secondary outcomes included an inter-group analysis of the patterns of the FOS+ neurons and the assessment of the differences between the EMG peculiarities during FW and BW stepping. A single animal was used as an experimental unit for all tests (n). For the EMG analysis, data are presented as median ± SD. Since there was a non-normal distribution of data (D’Agostino-Pearson test was used), a non-parametric Wilcoxon Signed Rank Test was used to determine the significance of the differences between individual pairs of means (FW vs. BW stepping). A p value of 0.05 was used as the cut-off for significance. Statistical calculations were performed using Prism 9.0 (GraphPad Software, La Jolla, CA, USA).

## 3. Results

*General pattern of rostrocaudal distribution of the FOS+ neurons over the lumbosacral enlargement.* All cats from the C-Fos group (6/6) were able to perform the long-lasted FW or BW locomotion and gave normal C-Fos labelling within the spinal cord grey matter. An unexpected non-uniform distribution of FOS+ neurons throughout the rostrocaudal axis (over the 40 slices, L1-S1 segments) was obtained. Independent from the direction of locomotion, two peaks in the number of FOS+ neurons were revealed (Figure 1B). These peaks had different expressions, possibly depending upon the locomotor mode. The first peak was located in segments L3–L4 in cat Fw1; in segments L4–L5 in cats Fw2 and Fw3; in segment L4 in cat Bw2; the peak was extremely weak in cat Bw3; and the peak was not visualized in cat Bw1. The second peak was located near the boundary between segments L6–L7 in cats Fw1 and Fw3; in segments L7-S1 in cats Fw2, Bw1 and Bw3; and in segments L6–L7 in cat Bw2. An inter-peak drop in the number of FOS+ neurons was visualized around the L5 and L6 segment boundary.

Note, that previously no clear interrelation between the level of the ES and the pattern of the FOS+ staining in a transverse plane was shown [13]. Similar evidence was obtained in the present work for the rostrocaudal distribution of the FOS+ neurons (red arrows in Figure 1B).

The exact location of the two peaks had high individual variability and an approximately 1 cm shift can be observed. This shift was more pronounced for the first peak. Our recent data give evidence for the variability of the segmental division of the lumbosacral spinal cord [23]; inter-animal dissimilarity in the rostrocaudal distribution of the FOS+ neurons was expected. A similar shift in the segmental location of the motoneuronal pools was obtained previously [17] and the authors elaborated on their own scheme for the definition of the lumbosacral spinal cord using easily recognizable features of the grey matter as landmarks. Thereby, during segmental analysis, we mainly relied upon the shape of the grey matter as mentioned above and correspondingly we shifted the curve of cat Fw1 caudally up to 5 slices.

Since there exist pronounced individual differences in the absolute numbers of the FOS+ neurons (an adverse event clearly dependent upon the multiple peculiarities of stepping), a percent amount of the labelled neurons was used instead of the absolute amount; and averaged curves for the rostrocaudal distribution of the FOS+ neurons were created (Figure 1C). Differences between these curves were revealed in two areas: (1) corresponding to the first peak in FOS+ neurons and (2) to the inter-peak drop in FOS+ neurons. The first peak and the inter-peak drop were both mainly developed in cats stepping FW (Figure 1C). The averaged curve in cats stepping BW was rather unimodal. To assess an inter-group difference between averaged curves, we at first transformed them into percent curves; thereafter we compared regions of the curves containing the first peak location and an inter-peaks drop (marked by red and gray dotted rectangles); a significant difference was obtained (*p* = 0.0078).

*Relationship between the distribution of the FOS+ interneurons and motoneuronal pools.* We had suggested that the rostrocaudal distribution of the FOS+ neurons generally corresponds to the distribution of the interneurons related to the different motoneuronal pools innervating the hindlimb muscles. To prove this, we compared the pattern of the FOS+ neurons and the distribution of different motoneuronal pools using the horseradish peroxidase (HRP) tracing data of Vanderhorst and Holstege [17], who presented a comprehensive study of hindlimb motoneuron tracing using 74 cats. We calculated labeled motoneurons at individual diagrams depicted in Figures 4, 7, 9, 11, 14, 16, 18 and 20 of [17] and assessed an averaged diagram for all hindlimb muscles. Thereafter we created a scheme of the rostrocaudal distribution of the different motoneuronal pools. As it can be seen from Figure 1C,D, the first peak in the FOS+ neurons is spatially related to the motoneuronal pools of the hip flexors and knee extensors (pink and red curves in Figure 1D); the second peak—to the plethora of motoneuronal pools related to the hip extension, knee flexion, abduction, ankle extension and ankle flexion—an inter-peak drop corresponds to the location of the motoneuronal pools of the adductors (green curve in Figure 1D) and partially, to the knee flexors, hip extensors, abductors and ankle extensors. The evidence for our supposition is a Pearson correlation coefficient between the angle range of the hip movements (allowing assessment of the general hip activity) and an expression of the first peak in FOS+ nuclei (a square of the first peak at the FOS+ curve illustrated at Figure 1B). Pearson coefficient was 0.924 (*p* = 0.0085).

*Peculiarities of the FOS+ neuron expression defined by the patterns of EMG activity.* To check whether the rostrocaudal distribution of the FOS+ neurons and the location of the motoneuronal pools are really coupled, in a separate experiment we analyzed the EMG activity of the muscles belonging to the three most rostral groups in the Vanderhorst and Holstege’s [17] scheme (hip flexors, knee extensors and adductors in Figure 1D). The IP was used as the demonstrative example of the hip flexors, the RF and VM as the knee extensors and the ADD as the adductors (Figure 2A) [24].

A predominance of the ADD during BW stepping and the IP, RF and VM during FW stepping were revealed in most cats (11/12 for ADD; 10/12 for IP; 6/6 for RF; 6/6 for VM). This can be clearly seen from both examples in Figure 2A. For all muscles, we analyzed an integrative spike of activity (envelope of the EMG burst activity during total step cycle) during two locomotor modes: FW and BW stepping. Despite inter-individual differences for absolute values of the EMG data, even averaged EMG amplitude for all muscles illustrated significant differences between FW and BW stepping (ADD: 0.0172 ± 0.0125 mV vs. 0.0489 ± 0.0448 mV, *p* = 0.0005, *n* = 12; IP: 0.0921 ± 0.0349 mV vs. 0.0599 ± 0.0286 mV, *p* = 0.0005, *n* = 12; RF: 0.1384 ± 0.0789 mV vs. 0.0232 ± 0.0278 mV, *p* = 0.0313, *n* = 6; VM: 0.0845 ± 0.0727 mV vs. 0.0207 ± 0.0083 mV, *p* = 0.0313, *n* = 6). To prove our data, we also created averaged percent curves (in relation to the whole step cycle) for the EMG signal during FW and BW stepping, where differences can also be seen for all muscles (*p* < 0.0001) (Figure 2B).

Also, to avoid a variability of absolute EMG values, we analyzed the percent ratio of the integrative spike activity between two locomotor modes evoked from the same ES point. Individual data for percent ratios is presented in Figure 2C; averaged data presented at Figure 2D. It can be seen that for the ADD muscles, most values were localized below the 100% point; for other muscles, values were localized above this point. The difference from 100% value is significant, for all muscles (IP: *p* = 0.0005; ADD: *p* = 0.0005; RF: *p* = 0.0313; VL: *p* = 0.0313). Higher EMG activity of the ADD during BW stepping is in-line with the higher step width (10/12 cats) during BW compared with FW stepping (201 ± 51 mm vs. 245 ± 72 mm, *p* = 0.0005, *n* = 12). Taken together, the lower level of ADD activity and the higher level of IP, RF and VM activity during FW stepping were in-line with more developed first FOS+ peak and inter-peak drop in the cats.

To test a possible link between the stability of the FW/BW stepping and an adductor activity, we at first performed an analysis of step width; it was significantly wider for BW locomotion (10/12; *p* = 0.0005) (Figure 3C). It means that during BW locomotion, hindlimbs are in a more abducted position than during FW stepping. Taking into account a less stability of the BW stepping in the sagittal plane (a percent ratio between stability for FW and BW stepping is less 100% (10/12; *p* = 0.0005) (Figure 3D), we performed testing with an additional abduction of hindlimb, in four cats (Figure 3A). An abduction led to the interesting changes in most muscles: (1) the magnitude of the ADD and IP was increased and RF—decreased, in all cats analyzed (4/4), an activity of the VL was changed arbitrarily. Due to the small sample, no significance was shown (ADD: *p* = 0.065; IP: *p* = 0.125; RF: *p* = 0.125). The ratio of the EMG amplitude during abducted FW stepping and FW stepping without abduction was 135 ± 13% for ADD, 111±14% for IP, 82.5±18% for RF and 110±34% for VM (Figure 3B). Thus despite little samples, a clear trend for three muscles to change their activity after abduction can be seen. More pronounced change was documented for ADD muscle.

## 4. Discussion

The overall motor output during locomotion depends upon the activity of the definite combinations of the hindlimb muscles. The distribution of motoneuronal pools controlling the muscles within the grey matter is highly ordered; their rostrocaudal positions reflect the proximodistal locations of the corresponding muscles [17,18]. Controlling locomotor networks consist of interneurons that exhibit highly structured spatial distribution correlated with the motoneuronal pools [25,26]; thereby, they can be visualized by the C-Fos technique allowing detection of the interneurons activated during different locomotor tasks [15].

Previously, the C-Fos technique was used for the mapping of the spinal locomotor networks in cats [13,15,16]. But the data from many slices of the same segment were summarized and no local rostrocaudal variations were presented. The independent analysis of the multiple slices in each segment over the lumbosacral enlargement performed in the present paper demonstrated that the rostrocaudal distribution pattern of the FOS+ neurons has two peaks: the first one was predominantly in the L4–L5 segments and the second in the L6-S1 segments, separated by the inter-peak drop in the L5–L6 segments. The magnitude of the first peak in the FOS+ neurons was higher in the cats stepping FW having higher EMG activity of the hip flexor and knee extensors. Such a specific difference in the EMG pattern in FW vs. BW locomotion corresponded to the increased angle range in these joints [13]; moreover positive correlation between the first FOS+ peak expression and the hip angle was observed. Based on this evidence, together with the spatial co-localization of the first FOS+ peak to the motoneuronal pools of the hip flexors and knee extensors [17], we propose that the interneurons attributed to the first peak are mainly related to the spinal circuits responsible for the control of hip flexion and knee extension. We did not perform a similar analysis for the second FOS+ peak because it corresponded to the location of the inter-mixed motoneuronal populations related to the numerous functional movements, including hip extension, knee flexion, abduction, ankle extension and ankle flexion. However, we believe that a more detailed analysis of the rostrocaudal distribution of the C-Fos stained neurons (using 20–30 slices of the lumbar segments) can give more precise data. Also, we believe that the presented quantitative combination of our own experimental data and experimental data in the literature can lead to the reduction of the use of animals in research.

The most striking result is a decrease in the number of FOS+ neurons within the area around the L5-L6 segments. According to Reference [17] scheme, this region contains motoneuronal pools of the adductor muscles. We revealed lesser activity of the ADD in cats stepping FW and a greater expression of the inter-peak drop in the cats. This is possibly related to lesser activity of the sensorimotor network controlling the adductor muscles in FW stepping cats. At first glance, it seems unusual; however, taking into account the specific body postural configuration of the cats with restrained hindquarters [27] and significantly wider steps during BW locomotion we can explain such a phenomenon. The hindlimbs during BW locomotion are in a more abducted position than during FW stepping, which seems to increase the positive feedback to the adductor muscles based on the stretch reflex mechanism [28]. To confirm it, we performed the test with additional abduction of the hindlimb during continuous FW stepping and revealed an increase of the adductor muscle amplitude in all animals tested after abduction.

It has been shown that walking is significantly dependent upon lateral body stability [29]. The 3D neuromechanical models of cat hindlimbs also have need for some imitation of the adductor/abductor muscles in controlling body stability in the frontal plane [30,31]. Moreover, in simulation of lateral stability, BW stepping requires a more complex pattern of generalized lateral force bursts then FW stepping does [31,32]. Adductors are involved in the postural stabilization and maintenance of the dynamic balance during locomotion [33]. Locomotion and postural control are tightly coupled; mechanisms participating in this integration are partially based upon the neuronal networks of the spinal cord and brainstem and are dependent upon sensory input [27,34,35,36]. BW locomotion has greater demands on postural control systems [5,37,38,39]. In the present paper, we obtained that BW stepping is less stable compared to FW stepping (10/12 cats) in terms of rhythmic movements in the sagittal plane (a percent ratio between stability for FW and BW stepping is presented in Figure 3D), which seems to require a wider support width in the frontal plane and more active adductor muscles to keep the center of mass closer and more stable in relation to the center of support [33,39].

BW and FW locomotion has different postural and hindlimb kinematics [40], kinetics [41] and muscle synergies [40,42] that require adaptive sensorimotor control carried out by the neuronal circuitry widespread over the spinal cord. In accordance, we have found a differently activated and functioning network distributed rostrocaudally in lumbosacral enlargement related to recorded properties of motor output during stepping in different directions. The characteristics of the C-Fos patterns observed here point out that slight variations in the gait and postural configuration are accompanied by explicit alterations in an expression pattern of the immediate early genes modulating the activity of the spinal sensorimotor networks. In practice, it is an additional direct evidence, confirming that training techniques can selectively recruit the specific neuronal network controlling particular motor functions. This is an objective argument based on the neuronal activity data toward the further modification of the motor rehabilitation protocols for multi-directional tasks [3,4,5] and involves the neuronal circuitry responsible for balance control and postural stability that is essential for patients with spinal cord injuries and other neuromotor disorders.

## Figures and Tables

**Figure 1 brainsci-11-00069-f001:**
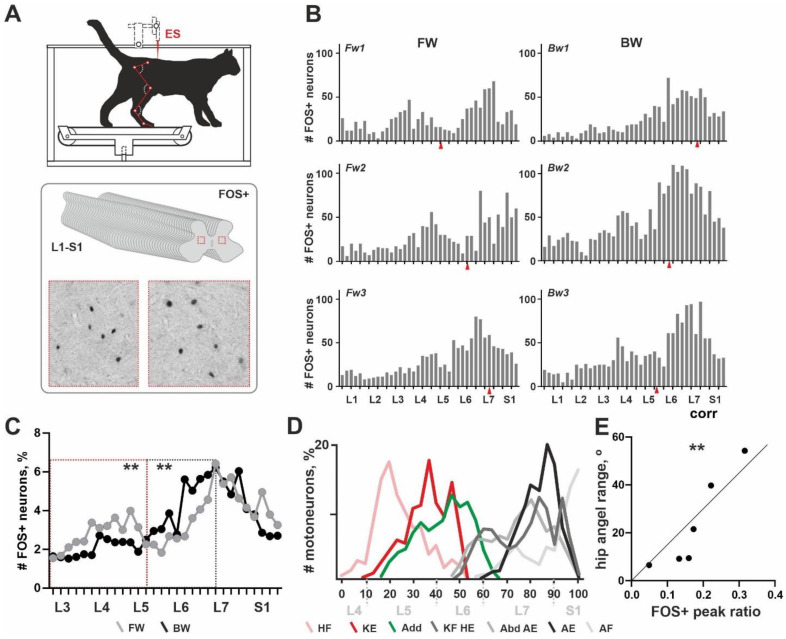
Rostrocaudal distribution of FOS+ neurons in cats stepping forward (FW) and backward (BW). (**A**) A scheme for the experiment. ES—epidural stimulation. Red boxes show examples of FOS+ neurons. (**B**) Individual data for individual FW (Fw1–Fw3) and BW (Bw1–Bw3) cats. OX axis, spinal segments, OY axis, absolute number of FOS+ neurons. Red arrows mark ES sites. (**C**) Difference between the averaged percent curves for the distribution of FOS+ neurons, for FW and BW cats; red and grey rectangles mark peak in the number of FOS+ neurons and inter-peak drop. (**D**) A rostrocaudal distribution of the motoneuronal pools assessed from the [17] data. (**E**) Pearson correlation coefficient between the angle range of the hip movements (allowing assessment of the general hip activity) and an expression of the first peak in FOS+ nuclei. HF—hip flexors; HE—hip extensors; KF—knee flexors; KE—knee extensors; AF—ankle flexors; AE—ankle extensors; Add—adductors. 0-100—rostrocaudal (R-C) levels of the spinal cord defined in Reference [17]. ** *p* < 0.01.

**Figure 2 brainsci-11-00069-f002:**
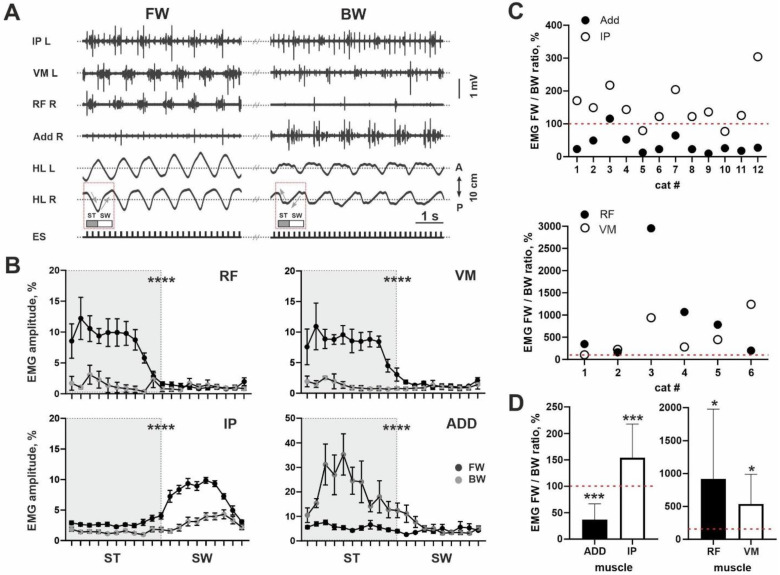
Muscle activity during FW and BW stepping. (**A**) Representative example of the EMG activity of the *m. iliopsoas* (IP), *m. rectus femoris* (RF), *m. vastus medialis* (VM) and *m. adductor magnus* (ADD) during the same registration. A, P—anterior and posterior directions of the hindlimb. SW—swing, ST—stance. HL L and HL R—left and right hindlimbs. ES—stimulation; light gray arrows mark hindlimb movements. (**B**) Averaged percent EMG activity during step cycle, for FW (dark gray) and BW (light gray) stepping. OX axis, step cycle. (**C**) Individual percent ratio in the muscle activity during FW and BW stepping, for ADD and IP (top) and RF and VM (bottom). OX axis, individual cats. (**D**) Averaged percent ratio in the muscle activity during FW and BW stepping. OX axis, muscles. * *p* < 0.05; *** *p* < 0.001; **** *p* < 0.0001.

**Figure 3 brainsci-11-00069-f003:**
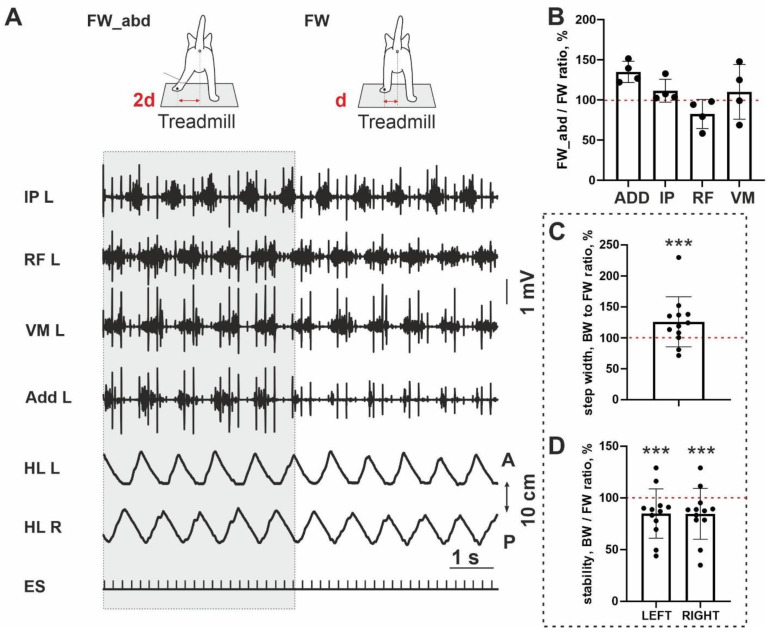
Abduction test and step characteristics during FW and BW stepping. (**A**) Representative example of the EMG activity during normal FW stepping (FW) and stepping with abduction (FW_abd). (**B**) The ratio in the muscle activity during FW and FW_abd conditions. OX axis, muscles. (**C**) A percent ratio of the step width between FW and BW stepping. (**D**) A percent ratio of the stability index between FW and BW stepping. Left, right—left and right hindlimb. *** *p* < 0.001.

## Data Availability

The data presented in this study are available on request from the corresponding author.

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
