# Peer review of "Rostrocaudal Distribution of the C-Fos-Immunopositive Spinal Network Defined by Muscle Activity during Locomotion"

_brainsci, 2021, doi:10.3390/brainsci11010069_

Round 1

Reviewer 1 Report

Article ”Rostrocaudal distribution of the c-fos-immunopositive spinal network defined by muscle  activity during locomotion” by Merkulyeva et al. compared interneuronal network activity in the lumbosacral level responsible for forward and backward stepping. Two activity peaks were revealed by positive c-Fos neurons: the first peak was in L4/l5 segments and the second peak was in L6/S1 segments. The first peak was correspondent to the motoneuron pool of the hip flexor and knee extensor in comparison with established data in cats. The key finding may be that the adductor muscles showed greater activity during backward locomotion, suggesting different locomotor networks for leg movements in different directions. Generally, the experiments were carefully designed and conducted, and the results were reliable based the data presented. My major concern is just the small sample number. There were only 3 cats in each group. The conclusion may be biased or misled by the small number of samples. I fully understand that cats cannot be used only for this kind of experiments. However, if the data can be validated further with larger sample size either with cats or other small animals, such as rats, it will be appreciated.

Minor concerns:

  1. It is not clear for how c-Fos positive neurons were counted and analyzed. Although from Figure 1A it seems only the central part of the lamina VII was used, the exact location and area of the analyzed region was not clearly stated. Also, a detailed description about the c-Fos positive neuron distribution in cross sections in different segments is lacking: were c-Fos neurons only located in the intermediate region or also in other regions, e.g., in the ventral horn?
  2. Page 1, line 2, in the title: “c-fos” needs to change to “c-Fos” because it is protein expression, not gene. Also change it in the main text wherever it is applicable.
  3. Page 4, Figure 1B, I suggest using the same y-axis scale for all cats so that the differences of c-Fos positive neurons between different cats were clearly visible.
  4. Page 5, lines 183-184, “The exact location of the two peaks had high individual variability, and a shift was observed near the total segment”. What does “the total segment” mean?
  5. Page 6, lines 195-197, “Differences between these curves were revealed in two areas: (1) corresponding to the first peak in FOS+ neurons and (2) to the inter-peak drop in FOS+ neurons”. Is there any statistical data to support this statement?
  6. Page 8, line 282, “……that during FW……”, “that” should be “than”.

Reviewer 2 Report

The authors demonstrate that forward and backward stepping - induced by epidural stimulation in decerebrated cats – at least partially activate different neuronal networks in the spinal cord. The muscle activity was investigated with EMGs from adductor, hip flexor and knee extensor muscles. The activated interneuronal networks was determined at different spinal segments using the c-fos technique. By using both of these measures, the association between the muscles activity and activated interneurons could be addressed for forward and backward stepping.

The main findings were that animals that underwent backward stepping had different rostrocaudal distribution of c-fos+ interneurons over the lumbosacral cord and different pattern of muscle activation than animals that underwent forward stepping.

The study design here is the same as a previously published paper from the same authors (Merkulyeva et al., 2018). The main differences between the present study and Merkulyeva et al. (2018) resides in the choice of muscles implanted with EMGs and the presentation of the c-fos analysis. Therefore, while the methodology used in this study is fairly strong, the novelty and the significance of the data are somewhat faint.

I have few comments and suggestions:

  • The use of a statistical tests showing significant group differences would greatly strengthen the data presented in figures 1C, 2B, 2C, 2D, 3B, 3C and 3D. Significant differences should also be indicated in figures (i.e. *p<0.05, **p<0.01, ***p<0.001).

  • The figure 1D shows the rostrocaudal distribution of the motoneuronal pools assessed from Vanderhorst and Holstege (1997). How the data was extracted should be clarified in the Methods section.

  • Some p values from figure 3 are clearly wrong (lines 286-287).

  • The results from figure 3 should be placed in the Results section, not in the Discussion.

Round 2

Reviewer 2 Report

I commend the authors for the extra analysis of the data and the revisions made. These changes have strengthened the data and the paper in general. I have only two minor comments.

  • In the Methods section (page 5), 5 animals are described to be used for the abduction test, as in the previous version of the manuscript. However, data for 4 animals are shown in figure 3B as well as in the corresponding Results section.

  • In the legend of figure 2 the statistical significance of p<0.05 is missing. This corresponds to the right part of figure 2D.
